# 1,6-Hexanediol Is Inducing Homologous Recombination by Releasing BLM from Assemblysomes in *Drosophila melanogaster*

**DOI:** 10.3390/ijms25031611

**Published:** 2024-01-28

**Authors:** Bence György Gombás, Zoltán Villányi

**Affiliations:** Department of Biochemistry and Molecular Biology, University of Szeged, H-6720 Szeged, Hungary

**Keywords:** assemblysome, 1,6-hexanediol, DmBLM, LLPS, homologous recombination

## Abstract

We recently demonstrated that 1,6-hexanediol inhibits the formation of assemblysomes. These membraneless cell organelles have important roles in co-translational protein complex assembly and also store halfway translated DNA damage response proteins for a timely stress response. Recognizing the therapeutic potential of 1,6-hexanediol in dismantling assemblysomes likely to be involved in chemo- or radiotherapy resistance of tumor cells, we initiated an investigation into the properties of 1,6-hexanediol. Our particular interest was to determine if this compound induces DNA double-strand breaks by releasing the BLM helicase. Its yeast ortholog Sgs1 was confirmed to be a component of assemblysomes. The BLM helicase induces DNA damage when overexpressed due to the DNA double-strand breaks it generates during its normal function to repair DNA damage sites. It is evident that storing Sgs1 helicase in assemblysomes is crucial to express the full-length functional protein only in the event of DNA damage. Alternatively, if we dissolve assemblysomes using 1,6-hexanediol, ribosome-nascent chain complexes might become targets of ribosome quality control. We explored these possibilities and found, through the Drosophila wing-spot test assay, that 1,6-hexanediol induces DNA double-strand breaks. Lethality connected to recombination events following 1,6-hexanediol treatment can be mitigated by inducing DNA double-strand breaks with X-ray. Additionally, we confirmed that SMC5 recruits DmBLM to DNA damage sites, as knocking it down abolishes the rescue effect of DNA double-strand breaks on 1,6-hexanediol-induced lethality in *Drosophila melanogaster*.

## 1. Introduction

We recently reported a significant discovery regarding the presence of DNA repair protein mRNAs in the cytoplasmic condensates of phase-separated ribosome nascent chain complexes (RNCs), referred to as assemblysomes [1,2]. It has been demonstrated that, in response to DNA damage, assemblysomes undergo a transformation into translating ribosomes that express DNA damage response proteins, such as the Sgs1 helicase [1]. Overexpression of BLM—which has functional similarities with DmBLM (in *Drosophila melanogaster*) and Sgs1 (in *Saccharomyces cerevisiae*)—is known to induce DNA damage [3,4,5,6,7,8,9]. Notably, BLM plays a crucial role in homologous recombination (HR) repair in eukaryotes [3,6,8,10,11,12].

The compound 1,6-hexanediol (HEX), a potent substance used to dissolve phase-separated organelles like P-bodies and stress granules (SGs), was found to reduce ring-oriented ribosomes reminiscent of assemblysomes in the A549 tumor cell line [1,13]. This reduction in parallel mitigated the DNA damage response in these otherwise radiation-resistant cells [1]. HEX has been shown to disturb liquid–liquid phase separation (LLPS) in the Drosophila singe cell system [14]. Furthermore, this chemical has been shown to increase general genome instability [15].

The dissolution of phase-separated ribosomes paused in translation by HEX raises intriguing questions about the fate of the RNCs released from assemblysomes. Do they become targets of ribosome quality control (RQC), recognizing stalled ribosomes to rescue them from depletion due to translation defects [16]? Conversely, do the condensates transform into translating ribosomes, completing the translation of potentially toxic gene products stored in assemblysomes until needed?

To investigate the impact of HEX on HR events in eukaryotes, we examined its effects in *Drosophila melanogaster* using the wing spot test assay [17,18,19]. The underlying principle of the wing spot test assay is as follows: the recessive *mwh* (multiple wing hairs) mutation, in its homozygous form, results in wing cells with 2–7 trichomes instead of one long hair [17,18,19,20,21]. To assess whether a mutagenic effect induces mitotic recombination, indicating chromosomal breaks, *mwh* heterozygous larvae were generated. These animals exhibit a wild-type phenotype, compensating for the *mwh* mutation’s lost function with functional genes inherited from one parent. Recombination during DNA repair may lead to a homozygous state on a chromosome, resulting in descendant cells expressing the characteristics of the established *mwh* homozygous state [17,19,22,23]. The Drosophila wing blade comprises approximately 30,000 cells, but even a single cell with the *mwh* phenotype is easily identifiable, standing out among the crowd of wild-type single wing hair cells (Figure 1) [20,21,22,23,24,25]. 

## 2. Results

### 2.1. 1,6-Hexanediol Is Less Toxic If Larvae Are Treated with X-ray

We subjected four larval groups, each comprising 50 larvae in biological triplicates, to the wing spot test assay. The four treatment groups included HEX, X-ray, a combined treatment of X-ray and HEX, and a non-treated control. It is important to note that we used F_1_ larvae originating from crossing *mwh*/*mwh* and *TM3,Ser* balancer carrier parents for assessing survival rates. Our results reveal that all three treatments significantly increased larval mortality compared to the control, and HEX alone produced the highest toxicity (Table 1, Figure 2A). In contrast, when double-strand DNA breaks (DSBs) were induced by X-ray radiation, HEX treatment showed less toxicity (Figure 2A) [25,26]. As a control, we treated wild-type larvae with 1,2-hexanediol and assessed survival rates. The phenomenon observed earlier with experiments performed with HEX is absent if larvae are treated with 1,2-hexanediol, a compound structurally similar to HEX but which has not been shown to have any effect on LLPS (Figure 2B).

While not statistically significant, the negative effect of 1,2-hexanediol on survival was additive to X-ray, unlike the effect of 1,6-hexanediol (Figure 2B). The variation in survival rates of X-ray-treated larvae between panels A and B in Figure 2 is attributed to the presence of the third balancer chromosome in experiments shown in A. Balancer chromosomes cannot function as templates for HR repair due to their inversions and chromosome rearrangements [20,27]. Therefore, it is expected to observe lower survival rates of X-ray treatment in crosses where one parent carries a balancer chromosome.

In the group treated exclusively with HEX, survivors likely belonged to a population highly resistant to HEX, as indicated by the relatively low number of mosaic patches suggesting DSBs (Table 1). Regarding the recombination frequency, calculated using the formula described in Materials and Methods, both the X-ray-treated and dual-treated groups exhibited rates two orders of magnitude higher than the control, with lower recombination frequencies observed in the X-ray-treated group. HEX alone also markedly induced HR events compared to the nontreated control, but the frequencies were considerably lower than those observed in the X-ray-treated groups (Table 1).

We observed an over-representation of balancer chromosome-carrying animals among the survivors in both HEX-treated groups (Figure 3). As no HR can occur on the balancer chromosome that contains multiple inversions, at least, on balancer chromosomes, HR-inducing DNA repair proteins released from assemblysomes are unable to exert their adverse effect, this explains the results shown in (Figure 3) [20,27]. The ratio of balancer-carrying survivals was significantly higher in both treatments where HEX was present, compared to either control or X-ray group, indicating the relation between HEX’s effect and HR. These results support the idea that DNA repair proteins involved in HR may be completing translation after their RNCs are released from assemblysomes due to HEX treatment.

### 2.2. Bloom Syndrome Helicase RNA Containing Assemblysomes Switches to Translating Ribosomes in Response to X-ray or HEX Treatment

We selected the Sgs1 ortholog DmBLM as a candidate assemblysome component and quantified DmBLM mRNA using qPCR associated with assemblysomes and polysomes. The preparation of assemblysomes and polysomes involved ultracentrifugation of larval extracts through a sucrose cushion, following the procedure outlined in [1,2].

In brief, assemblysomes exhibit resistance to ethylenediaminetetraacetic acid (EDTA), enabling them to pass through a 60% sucrose cushion even in the presence of EDTA, which typically disconnects the small and large ribosome subunits, leading to mRNA accumulation at the top of the cushion [1,2]. All extracts were divided into two equal volumes and pelleted through sucrose cushions in the presence of EDTA (for assemblysomes) or in the absence of EDTA (for polysomes). Larvae were either treated or left untreated with X-ray or HEX.

Expressing the results as the pellet/EDTA-pellet DmBLM mRNA ratio allowed us to monitor how DmBLM switches between assemblysomes and polysomes in response to X-ray and HEX treatments (Figure 4). Both X-ray and HEX treatments caused DmBLM mRNA to shift towards EDTA-sensitive ribosomes, evident from the increased pellet/EDTA-pellet DmBLM mRNA ratio in response to both treatments. Notably, HEX had a more pronounced effect, increasing the pellet/EDTA-pellet ratio of DmBLM mRNA approximately 4 times compared to the control, while X-ray increased it about 2.5 times. This suggests that X-ray has an impact on only a relatively small proportion of the available BLM-containing assemblysome pool compared to HEX.

### 2.3. The SMC5/6 Complex Is Involved in the Recruitment of Assemblysome-Regulated DNA Repair Machinery to the Site of DNA Damage

The intriguing observation that the detrimental impact of HEX on Drosophila survival can be alleviated by X-ray treatment led to the hypothesis that enhanced survival rates result from a factor of recruiting DNA repair proteins released from assemblysomes to the site of damage. Consequently, it appears advantageous for DNA repair proteins, such as BLM, to be released from assemblysomes just before X-ray treatment induced by HEX compared to the group treated only with HEX. Confirming the presence of BLM in assemblysomes in *Drosophila melanogaster*, we directed our attention to the SMC5/6 complex, known for mediating/recruiting RecQ helicases in eukaryotes (e.g., yeast Sgs1) to DSBs [28,29,30,31,32,33]. 

We generated SMC5-RNAi-expressing Drosophila using the UAS Gal4 system and conducted survival tests on the same four treatment groups of larvae, employing ubiquitously expressed SMC5-RNAi animals with the aid of the daughterless-Gal4 driver. In SMC5-RNAi flies, the detrimental effect of HEX was additive to X-ray treatment, resulting in the lowest survival rates in the HEX and X-ray double-treated group among the four studied treatment groups (Figure 5). Importantly, in the case of silenced SMC5, double-treatment resulted in significantly lower survival than HEX or X-ray alone, which highlights the additive effect of the two, unlike the original experiment, where double-treated animals produced a higher survival rate than HEX alone.

This outcome suggests that if BLM, and possibly other DNA repair components, are not efficiently recruited to the site of DNA damage, the excess DNA repair proteins released from assemblysomes due to HEX may initiate recombination events randomly, not necessarily at the site of X-ray-induced DNA DSBs. This manifests as an additive effect of the two treatments, emphasizing not only the presumed role of the SMC5/6 complex in recruiting BLM to DNA DSBs but also highlighting the significance of assemblysomes in HR repair. Both the driver daughterless-gal4 and UAS-SMC5-RNAi flies were homozygotes for their respective transgene; therefore, all F_1_ offspring were SMC5 silenced and did not contain balancer chromosomes. But even comparing the results to an experiment where balancer chromosomes were present and reduced survival rates of solely X-ray-treated larvae, SMC5-silenced Drosophila show hypersensitivity to genotoxic radiation as expected (Figure 2A). This confirms that the Drosophila stock used in this study potently silenced SMC5 as reported earlier (Figure 5) [34].

### 2.4. UV-Induced DNA Damage Is Additive to HEX’ Effect on Mortality

UV radiation causes DNA damage, specifically affecting only one DNA strand; therefore, this damage is not corrected through HR but rather through nucleotide excision repair [35]. Given that UV radiation is unlikely to induce HR, we conducted the original experiment using UV radiation instead of X-rays. The aim was to determine if a radiation type causing DNA damage not necessarily dependent on DmBLM for repair could still produce a rescue effect on HEX, similar to X-rays. This would indicate that HEX is acting on another pathway in generating genomic instability and not by acting on the LLPS of assemblysomes through liberating BLM. Our findings revealed that UV irradiation did not have a rescue effect on HEX. Additionally, we observed an additive toxicity when both UV and HEX treatments were applied to Drosophila (Figure 6). These results lead us to the conclusion that the mechanism induced by HEX is intricately linked to HR.

## 3. Discussion

Our findings reveal that at least DmBLM, the ortholog of yeast Sgs1, is stored in LLPS assemblysomes and undergoes a transition to translating ribosomes upon treatment with HEX in third instar larvae of *Drosophila melanogaster*. This unique property of HEX stems from its ability to dissolve phase-separated membraneless organelles formed by LLPS, such as assemblysomes known to store partly synthesized DNA repair proteins, including DmBLM [1]. This transition coincides with a significant increase in recombination events measured on the wings (Table 1). These results strongly suggest that HEX dissolves assemblysomes, allowing paused ribosomes to resume translation on the mRNA template, complete the translation process, and produce functional proteins that induce recombination events. Importantly, these RNCs, after liberation from assemblysomes, escape RQC. It remains elusive if this is due to exhausting the RQC system with the sudden appearance of a plethora of paused ribosomes, or if the pause in translation is easily lifted after phase transition of RNCs triggered by HEX. The former scenario is plausible according to our results in Figure 4 showing that HEX releases much more DmBLM mRNA than X-ray radiation from assemblysomes, representing a sudden high burden on translation quality control processes. 

Furthermore, our findings highlight that the toxicity of 1,6-hexanediol is contingent on the pool of assemblysomes present in the cell. Cells exposed to substantial amounts of genotoxic compounds, such as those found in cigarette smoke, may have a higher abundance of DNA repair components midway through synthesis in assemblysomes compared to non-stressed cells [36]. Considering that tumor cells often originate from stressed cells undergoing oncogenic transformation, HEX emerges as a potential agent of interest. It has the capacity to induce more recombination events in malignant cells than in healthy ones due to the elevated number of assemblysomes containing gene products expressing DNA repair proteins in the former [1,37]. Additionally, radioresistant tumors may exhibit resistance to therapy due to an increased accumulation of DNA repair components stored in assemblysomes. 

We posit that HEX could serve as a potent chemotherapeutic agent for the treatment of cancers originating from the malignant transformation of healthy cells, particularly those influenced by exposure to toxic environmental perturbations. The intricate interplay between HEX, assemblysomes, and genomic instability opens avenues for further exploration in cancer research and therapeutic development.

## 4. Materials and Methods

### 4.1. Genotypes, Markers, Crosses, and Strains

For the somatic mutation and recombination test (SMART), two Drosophila strains were utilized: Bloomington Drosophila Stock Center’s 2371 (*flr/TM3,Ser*) and 549 (*mwh/mwh*) stocks [19,38]. The *mwh* (multiple wing hairs) recessive allele, located on the left arm of chromosome 3, induces the growth of additional trichomes on the wing blades compared to the wild-type [19,20,21,22]. The *Ser* (*Serrate*) dominant marker produces a noticeable wing indentation, and the balancer chromosome *TM3* inhibits HR during the experiment [19,20,27,38]. Marker heterozygous larvae were created by crossing the two strains in both directions. The animals were stored at 25 °C on standard Drosophila medium until the treatments.

In the case of 1,2-hexanediol treatment and UV irradiation, wild-type *Drosophila melanogaster* strains were used.

Silencing of SMC5 using siRNA in the UAS Gal4 system involved crossing Bloomington Drosophila Stock Center’s 56035 with the daughterless Gal4 driver line (that allows expression of the specific interfering RNA in every tissue, in every developmental stage, which is important for successful silencing in imaginal discs of the third instar larvae). Both parents were homozygotes for the respective UAS Gal4 components they carried. Third instar larvae from generation F_1_ were used for treatments detailed below in the “Larval Treatments” chapter.

### 4.2. Somatic Mutation and Recombination Test (SMART)

The SMART wingspot test, largely adopted from [17,18,19,38], evaluates the mutagenic/DNA-modifying capacity of various treatments in vivo. The method relies on somatic mutations manifested as mosaic spots in adult fruit fly wings. The formula ($f=\frac{nm}{NC}$; where “n” is the number of mosaic spots, “m” is the average size of the spots, “N” is the number of wings checked, and “C” is the number of cells comprising a wing blade (about 30,000)) helps infer mutational events in larval imaginal discs [19,21,38]. Larvae were treated, collected, and raised in test tubes until they developed pupae. The wings of hatched adult animals were first prepared in Faure’s solution (to make them examinable by microscopy), then checked for *mwh* mosaic spots to assess mutagenic properties. We also determined the frequency of mosaic spots ($\frac{n}{N}$). The grown flies after treatment were stored at 70% ethanol until the preparation of the wings. Wings were observed under a light microscope (Olympus CX21) at a magnification of 400×. Cells were counted manually to calculate the size of the spots (each scorable unit = 1 cell [21]). 

### 4.3. Larval Treatments

Once 90–94 h old, F_1_ generational larvae of specific crosses were divided into four groups (50–50 in all three biological replicates): control, HEX, X-ray, and HEX + X-ray. HEX treatment (30% *v*/*v*) and X-ray treatment (with the dose of 1000 Rad) were administered separately or in combination. The treatment was executed in test solution (50% *v*/*v* standard Drosophila medium mixed with: 20% *v*/*v* distilled water and 30% *v*/*v* HEX; or 50% *v*/*v* distilled water) for 1 h. The fact that the treatment happened in a viscous liquid state of matter prevented the larvae from climbing up from the chemical. The presence of Drosophila medium prompted the animals to consume HEX dissolved in the solution. After larvae spent 1 h in test solution, we placed them in Petri dishes filled with 14% NaCl solution that made them remain located in one plane on the surface of the solution; therefore, the animals were irradiated equally by X-ray and UV (performed immediately after HEX treatment). X-ray treatment was executed with a TRAKIS XR-11 machine. After irradiation, we placed the larvae back on standard Drosophila medium and raised them at 25 °C until they formed pupae and hatched. Survival tests were performed by counting successfully hatched adult flies. 

Similar to HEX, 1,2-hexanediol treatment (15% *v*/*v*) followed the same procedure in test solution. 

UV irradiation (18 mJ/cm^2^, 2 min) was performed in a completely similar way to X-ray treatment in 14% NaCl solution.

All larval experiments were repeated 3 times with independent biological replicates performed on 50 synchronized larvae.

### 4.4. Isolation of Assemblysomes

The mechanical homogenization of Drosophila tissue was executed in lysis buffer (300 µL per sample) using a plastic mortar that fits 1.5 mL tubes. We lysed cells by vortexing tubes containing the homogenized tissue for 15 min at 4 °C in the presence of 0.5 mL glass beads with diameters between 425 µm and 600 µm (SIGMA, Budapest, Hungary) in each sample [1]. After lysates were pulled off the beads by centrifugation through a short spin, we executed an 8000× *g* centrifugation for 10 min to get rid of protein aggregates, nuclei, SGs, and cell debris [1]. Assemblysomes and polysomes were isolated by ultracentrifugation of cell lysates on a 60% sucrose gradient with or without EDTA (25 mM final cc.) following the workflow in [1]. Samples were ultracentrifuged for 4 h at 50,000 rpm at 4 °C in a Sorvall MX 120/150 Plus Micro-Ultracentrifuge (Thermo Fisher Scientific, Budapest, Hungary) in S55A2 rotor. Pellets from ultracentrifugation were resolubilized in lysis buffer for RNA examination. In the absence of EDTA, the pellet is enriched with polysomes and assemblysomes, while adding EDTA leads to enrichment of assemblysomes only [1,2]. Supernatant was pipetted off from the pellet which was used for further procedures described below. The whole procedure was conducted at 4 °C.

### 4.5. RNA Extraction—Quantitative Real-Time Polymerase Chain Reaction

RNA extraction, reverse transcription (RT), and qPCR followed the workflow from [1]. After ultracentrifugating the total extracts of the treated Drosophila larvae (in the presence or absence of 25 mM final concentration EDTA), we isolated RNA from the pellets using TriTrep Kit (Macherey-Nagel, Oensingen, Switzerland). After measuring the RNA concentrations with a NanoDrop spectrophotometer, we reverse transcribed 0.1 ng–5 µg of it with random hexamer primers of RevertAid First Strand cDNA Synthesis Kit (Thermo Fischer Scientific), then diluted cDNA were used for quantitative real-time qPCR with the GoTaq qPCR Master Mix (Promega, Vienna, Austria). For analysis, we used Piko-Real 96 Real-Time PCR System (Thermo Fischer Scientific), and gene specific primers were used to detect DmBLM and TUB1A1, which was used as a loading control. Pellet/EDTA–pellet ratios were calculated. We conducted the measurements of two independent biological replicates in each case.

### 4.6. Chemicals

1,6-Hexanediol (SIGMA): Used for inhibiting liquid–liquid phase separation. A 60% *v*/*v* stock solution was diluted to 30% *v*/*v*. 

1,2-Hexanediol (SIGMA): control molecule with a similar structure to HEX, diluted to a 60% *v*/*v* stock solution, used in test solution (15% *v*/*v*).

Faure’s Solution: used as a mounting medium for wings, composed of 50 g chloral hydrate, 50 mL distilled water, 30 g gum arabic, and 20 mL glycerol.

Lysis buffer (used for lysis of Drosophila cells and larval tissue homogenization): 100 mM KCl, 50 mM Tris-Cl pH 7.4, 1.5 mM MgCl_2,_ 1 mM DTT, 1.5% *v*/*v* NP-40, Protease Inhibitor Cocktail (Roche, Germany), and Ribolock RNase Inhibitor (Thermo Fischer Scientific, Vilnius, Lithuania).

Glass beads (SIGMA, 425–600 µm): tool for mechanical cell lysis of treated larvae.

## 5. Conclusions

This study unveils a key finding: assemblysomes, harboring HR elements temporarily halted in translation with the ability to resume translation upon encountering DNA damage for effective repair, are significantly influenced by HEX. Notably, HEX exhibits an additional level of toxicity in comparison to its counterpart, 1,2-hexanediol. This heightened impact is a direct result of HEX’s influence on LLPS, affecting—among other cellular processes—the dynamics of assemblysomes. Consequently, there is a notable release of DmBLM and possibly other HR components in sudden and elevated concentrations, evading the regulatory mechanisms of translation quality control. In essence, HEX’s influence on assemblysomes triggers the release of HR components in an uncontrolled manner, leading to the initiation of detrimental HR events across the genome without the necessity of recruitment to specific sites in the absence of DNA damage. This positions HEX as a potential compound to instigate genome instability, particularly in cells that have accumulated assemblysomes, such as cancer cells resulting from malignant transformation induced by DNA-damaging agents.

## Figures and Tables

**Figure 1 ijms-25-01611-f001:**
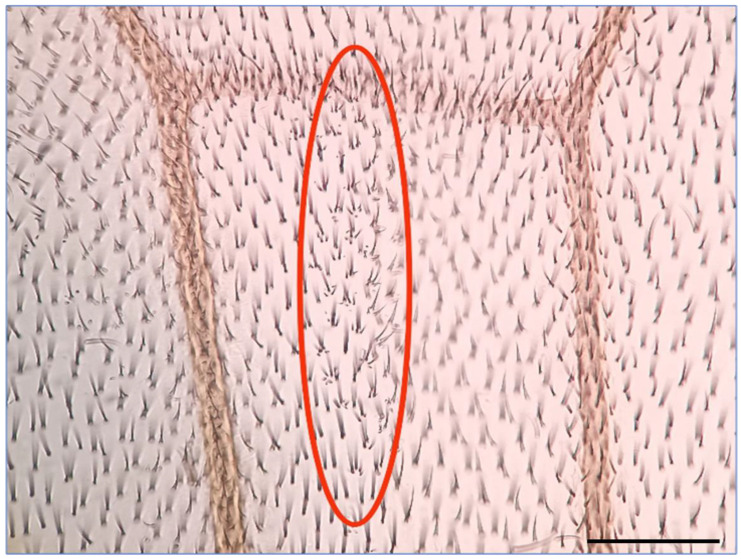
A typical *mwh* spot on the wing of *mwh/+* heterozygote *Drosophila melanogaster* is an indicator of a HR event. The *mwh* spot is circled in red. Scale bar: 50 µm.

**Figure 2 ijms-25-01611-f002:**
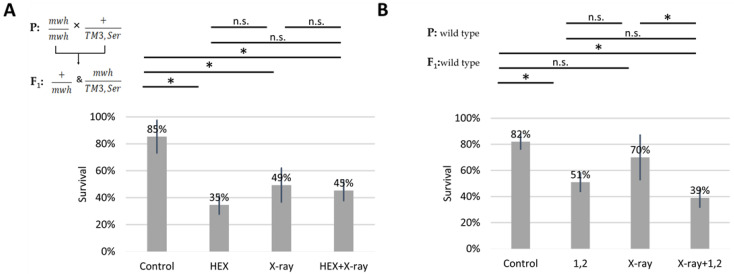
Effect of different hexanediols and genotoxic radiations on survival of *Drosophila melanogaster*. Survival is expressed in percent in the four indicated treatment groups showed on the diagram. On (**A**): HEX: 1,6-hexandiol and on (**B**): 1,2: 1,2-hexanediol was used to treat the larvae. The results are a summary of three independent measurements (n = 50). Parental (P) and F_1_ genotypes are shown in the top left corner for A and B. F_1_ larvae were treated as indicated. Statistical analysis was performed with Student’s *t*-test and the results are highlighted above the diagram (* *p* < 0.05, n.s.: non-significant).

**Figure 3 ijms-25-01611-f003:**
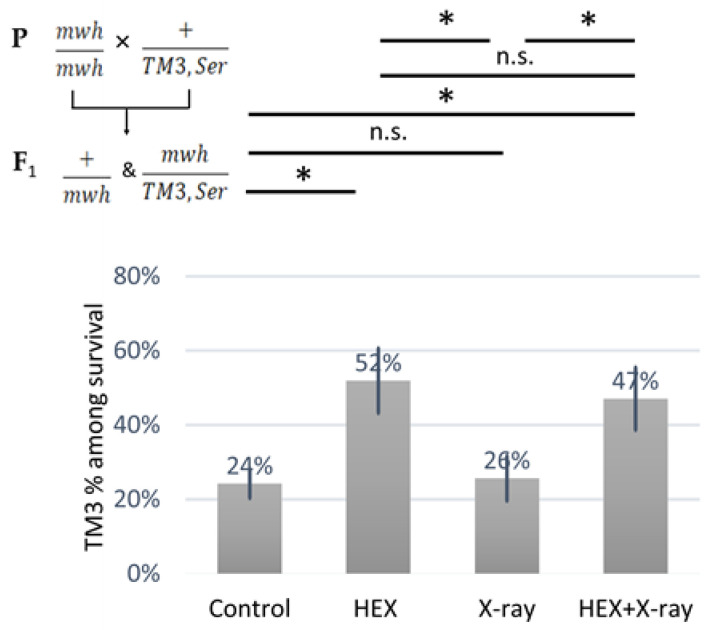
Animals carrying a *TM3,Ser* balancer chromosome resist the deleterious effect of HEX. The diagram shows the percent of the animals carrying the balancer chromosome *TM3* among the survivals. The results are a summary of three independent measurements (n = 50). Parental (P:) and F_1_ genotypes are shown in the top left corner. F_1_ larvae were treated as indicated. Statistical analysis was performed with Student’s t-test and the results are highlighted above the diagram (* *p* < 0.05, n.s.: non-significant).

**Figure 4 ijms-25-01611-f004:**
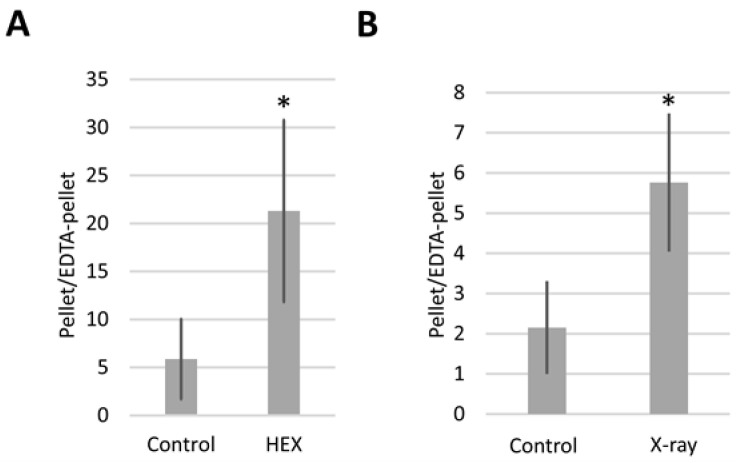
pellet/EDTA–pellet ratio of BLM mRNA as measured by qPCR. In the case of (**A**), larvae were treated with HEX; in the case of (**B**), with X-ray. The figure represents the average of two independent measurements of biological replicates. Statistical analysis was performed with Student’s *t*-test (* *p* < 0.05).

**Figure 5 ijms-25-01611-f005:**
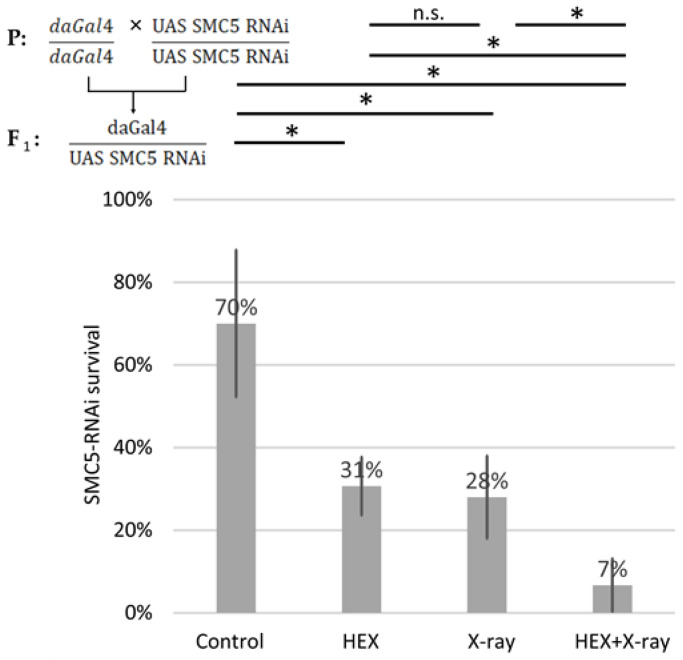
SMC5 silencing suppresses the rescue effect of X-ray on lethality induced by HEX. Survival is expressed in percent in the four indicated treatment groups showed on the diagram. The results are a summary of three independent measurements (n = 50). Parental (P) and F_1_ genotypes are shown in the top left corner. F_1_ larvae were treated as indicated. Statistical analysis was performed with Student’s *t*-test and the results are highlighted above the diagram (* *p* < 0.05, n.s.: non-significant).

**Figure 6 ijms-25-01611-f006:**
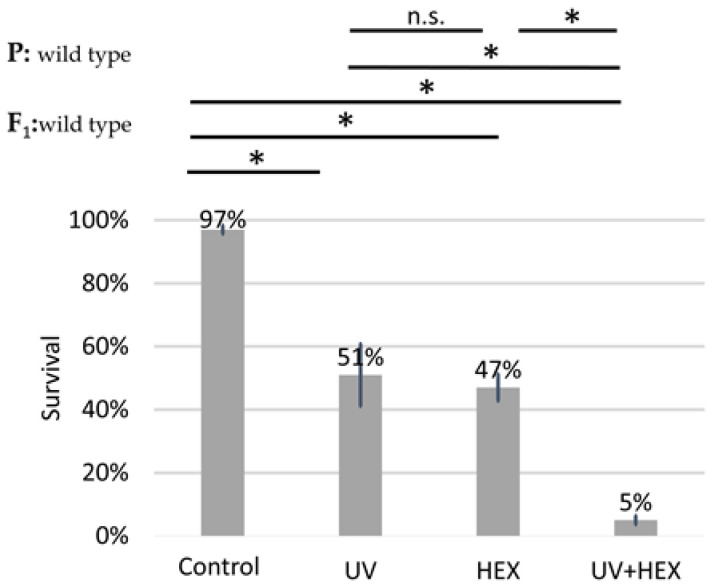
Unlike X-ray, UV-induced damage is additive to HEX’ effect on mortality. Survival is expressed in percent in the four indicated treatment groups showed on the diagram. The results are a summary of three independent experiments (n = 50). Parental (P:) and F_1_ genotypes are shown in the top left corner. F_1_ larvae were treated as indicated. Statistical analysis was performed with Student’s t-test and the results are highlighted above the diagram (* *p* < 0.05, n.s.: non-significant).

**Table 1 ijms-25-01611-t001:** Results of the *Drosophila melanogaster* wing spot test assay. The frequency of recombination events and the strength of the mutagenic effect can be measured with the formulas described in Materials and Methods [19,21,22].

	Control	HEX	X-ray Treatment	HEX+X-ray Treatment
Number of treated larvae	150	150	150	150
Number of mosaic spots (*mwh*) between mutants	2	4	25	40
Number of *mwh/TM3,Ser* survivals	31	27	19	32
Number of *mwh/+* survivals	97	25	55	36
Number of examined wing pairs (all survivals)	128	52	74	68
Average size of *mwh* spots in trans-heterozygotes ± standard deviation	3 ± 0	4.5 ± 3.4	27.6 ± 43.1	16.5 ± 30.9
Frequency of *mwh* spots (*mwh*)	0.0103	0.08	0.2273	0.5556
Recombination frequency	1.0309 × 10^−6^	1.0667 × 10^−5^	2.0939 × 10^−4^	3.0602 × 10^−4^

## Data Availability

The data that support the findings of this study are available from the corresponding authors upon reasonable request.

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
