# Peer review of "1,6-Hexanediol Is Inducing Homologous Recombination by Releasing BLM from Assemblysomes in Drosophila melanogaster"

_ijms, 2024, doi:10.3390/ijms25031611_

Round 1

Reviewer 1 Report

Comments and Suggestions for Authors

This manuscript (MS) deals with the effects of 1,6-hexanediol (HEX) on survival and homologous recombination (HR) level in D. melanogaster. The authors ascribe the toxic effect of HEX on survival to increased HR, and specifically to increased level of BLM helicase in those cells. Although the topic is quite interesting, there are some problems concerning the MS.

1)     The effect of HEX on viability is quite low and the combination with X-rays is not statistically different. If the tumor cells are considered similarly stressed as the ones irradiated with X rays, the application of HEX would similarly have no meaningful effect on them, hence the HEX perspective for cancer cure is likely limited.

2)     The explanation that HEX induces dissolution of assemblysome with ensuing release of BLM, which then reduces viability by acting on DNA is interesting and is partially supported by the data. However, the data in Fig 5 argue against this explanation. Namely, the mutants with inactivated BLM recruiter to damaged DNA, SMC5, still have the decreased viability upon HEX application, to the similar level as in wt animals. This suggests that the reduced viability in HEX treated animals is unlikely due to BLM activity on DNA.

Moreover, the survival of X-irradiated cells with inactivated SMC5 are about 30%, while their wt analogs survive at about 50% (compare Fig 2 and Fig 5). Is BLM helicase so unimportant for X-ray damage repair that its inability to act on damaged DNA causes less than two-fold reduction in viability?! Or, alternatively, the SMC5 silencing was somewhat inefficient? Is there any control to show the efficiency of SMC5 silencing?

3)     It is not clear from the MS what was the sequence of application of HEX and X (UV) rays, which is important. The procedures should be described in much more detail.

4)     The data are shown in Fig. 6, which are not described in the Results nor Discussion sections.

5)     The numbers of figures are not correct: line 124 should refer to Fig. 4, and Fig 4 in line 144 should be changed to Fig. 5. Also, the Fig 2 should be presented the first in the Results section, while the Table 1 second, since this is the sequence of the described data.

Comments on the Quality of English Language

Please use decimal point instead of comma. Also, there are too many decimals in the fractional part of the number, which reduces clarity of data.

Reviewer 2 Report

Comments and Suggestions for Authors

Gombas and Villanyi present a study that explores an exciting new feature of DNA repair regulation using an ingenious combination of classical genetics and modern biochemistry. Their study is an extension of previous work from the Villanyi lab where they were able to show that ribosomes translating proteins with disordered N-termini and ubiquitination sites have the ability to form assemblysomes, i.e. LLPS-like aggregates that are blocked in translation, but can be activated upon stimuli such as e.g. DNA damage. In this paper, they use the powerful Drosophila genetics approach to investigate assemblysomes containing BLM helicase. They find that exposure to 1,6-hexane diol is toxic for Drosophila larvae and that the toxicity is attenuated by X-ray irradiation, an effect not observed when they use 1,2-hexane diol, which is not able to disrupt LLPS. They find that BLM is released from assemblysomes both due to 1,6-hexane diol treatment and in response to irradiation. They observe increased homologous recombination in response to both 1,6-hexane diol and irradiation. The authors conclude that release of BLM by 1,6-hexane diol treatment frees BLM, leading to spurious HR initiation the damage from which is attenuated by concurrent induction of DNA lesions due to irradiation. 

The authors extend experimental techniques that have been established for single cell systems for the use in a complex organism, Drosophila third instar larva, without providing evidence of the validity of the approach. If the authors can show that their treatment with 1,6-hexane diol disrupts LLPS in the entire larva (or at least in the imaginal disks), this paper very, very much deserves publication. I am very enthusiastic about the paper and the technique described, which, if valid, would allow to study the effects of LLPS disruption in metazoan organisms, especially with the ability to harness the power of Drosophila genetics.

The paper is very well written, clearly structured and does not require additional language editing.

Major issues:

This report is to my knowledge the first using 1,6-hexane diol-mediated LLPS disruption in entire metazoans (Drosophila third instar larvae) as opposed to single celled organisms or metazoan cells in cell culture. It has been shown that 1,6-hexane diol efficiently disrupts LLPS inside yeast and human cells by fluorescence microscopy. While there is no reason to assume that it would not work in Drosophila (and it has been shown to work in Drosophila cells, e.g. Amankwaa et al., 2022), a control showing that LLPS is disrupted by incubation in 1,6-hexane diol in the Drosophila larva (e.g. in the wing disk) would be essential to demonstrate the validity of the approach as it is not clear whether the reagent is able to penetrate the entire third instar larva to reach the wing disc under the conditions used in the experiment. This control could e.g. be accomplished using a LLPS-sequestered fluorescently labeled protein with live imaging of the entire larva. Several publications have remarked on the substantial toxicity of 1,6-hexane diol to e.g. yeasts (e.g. Jones and Forsburg, 2023). One could imagine that Drosophila larvae might be a bit harder to penetrate for the compound due to their cuticle and their overall much larger mass. So, it would be important to show that LLPS dissolution really does take place in the larva, possibly combined with a titration of 1,6-hexane diol concentration.

It has previously been shown that 1,6-hexane diol is a rather blunt instrument, causing widespread disruption to cellular processes. In S. pombe, genome stability was affected by 1,6-hexane diol incubation (Jones and Forsburg, 2023). The authors suggest a very specific mechanism for the observed increase of homologous recombination upon 1,6-hexane diol treatment, BLM liberation from LLPS assemblysomes (which they demonstrate convincingly), but can not exclude that genomic instability originated from other pathway disruptions. In BLM KO flies (McVey et al., 2007), irradiation should not attenuate 1,6-hexane diol toxicity.

Minor issues:

There are several issues with the presentation: The authors provide a summary of their experimental results in table 1 where they list the overall spot numbers etc. for all flies and then for flies containing the balancer chromosome. This is a bit confusingly presented and a redesign of the table could really help. Average and SD could be presented in the same row to make it more concise.

Plot axes should be labeled.

Units given in the table are confusing. Is the average size of spots really in meters? The cell number in the wing is a value taken from the literature, it probably does not need to be presented four times in the table and could be included in the formula instead.

How many digits do the authors consider significant? Is it really necessary to present e.g. the frequency of spots to the ninth digit? Also, the authors should use correct mathematical typesetting to improve the presentation of formulae and the rec. frequency; MDPI layout uses LaTeX, so this can definitively be done and will improve the presentation substantially.

Round 2

Reviewer 1 Report

Comments and Suggestions for Authors

The manuscript is sufficiently improved to make it suitable for publication in IJMS.

Author Response

Once again, we express our gratitude for the constructive comments of Reviewer 1, and for finding our MS suitable for publication.

Kind regards,

Zoltan Villanyi

Reviewer 2 Report

Comments and Suggestions for Authors

I would like to thank the authors for their effort in improving the paper and addressing the concerns voiced in my review. As stated in the first review, their paper investigates the genetic consequences of chemical LLPS disruption in third instar Drosophila larvae and introduces this entirely novel experimental system with reference to previous work on single celled organisms and cultured cells, but without direct experimental evidence of successful LLPS disruption in larvae. I agree with the authors that a full investigation of HEX uptake and permeation in the larvae is a substantial undertaking. Although quick and simple controls could be devised, e.g. lacing the HEX medium with a coloring agent to see whether it is actively ingested by the larvae (if something like this is possible), actually showing disruption of LLPS inside the entire larva involves sophisticated imaging and is a (very promising and meritorious in my opinion) project in itself. Especially if the authors are, as I hope, planning to utilise this system for further investigation, a better characterisation would be very important. I agree with the authors that disruption of BLM assemblysomes in HEX treated larvae can be regarded as a biochemical proxy to observation of LLPS disruption and I do not want to place undue burden on them, especially as they present several lines of evidence that support their conclusion.

The authors have substantially improved the text, especially the description of the genetic experiments is now much clearer. The figures and the table have been improved; it's great to include the fly genetics crossing schemes in the figures! The materials and methods section now contains a better description of the HEX ingestion experiment.

Typo:

333: distilled

In conclusion, I congratulate the authors on this exciting study and fully recommend publication.

Author Response

We are pleased to inform Reviewer 2 that we are actively considering further experiments and characterizations, and her/his encouragement in this regard is both motivating and appreciated.

We are particularly grateful for her/his positive feedback on the improved clarity of the genetic experiments, figures, and table. 

Also, thanks for the reviewer for pointing out the typo in line 333; we corrected it.

Once again, we express our gratitude for the thorough review of Reviewer 2 and for the endorsement of our study.

Kind regards,

Zoltan Villanyi